# Biomarkers for Prostate Cancer: From Diagnosis to Treatment

**DOI:** 10.3390/diagnostics13213350

**Published:** 2023-10-31

**Authors:** Jia-Yan Chen, Pei-Yan Wang, Ming-Zhu Liu, Feng Lyu, Ming-Wei Ma, Xue-Ying Ren, Xian-Shu Gao

**Affiliations:** 1Department of Radiation Oncology, Peking University First Hospital, Beijing 100034, China; chenjiayan2019-ro@bjmu.edu.cn (J.-Y.C.); lyufeng2018@bjmu.edu.cn (F.L.); dr.mingweima@stu.pku.edu.cn (M.-W.M.); xy_ren1031@126.com (X.-Y.R.); 2School of Information, University of Michigan, Ann Arbor, MI 48109, USA; peiyanw@umich.edu; 3Clinical Oncology School of Fujian Medical University, Fujian Cancer Hospital, Fuzhou 350014, China; liumingzhu125@126.com

**Keywords:** prostate cancer, biomarkers, PHI, 4K score, PCA3, Decipher, Prolaris, BRCA1/BRCA2, ETS gene fusions, disease management, precision therapeutics

## Abstract

Prostate cancer (PCa) is a widespread malignancy with global significance, which substantially affects cancer-related mortality. Its spectrum varies widely, from slow-progressing cases to aggressive or even lethal forms. Effective patient stratification into risk groups is crucial to therapeutic decisions and clinical trials. This review examines a wide range of diagnostic and prognostic biomarkers, several of which are integrated into clinical guidelines, such as the PHI, the 4K score, PCA3, Decipher, and Prolaris. It also explores the emergence of novel biomarkers supported by robust preclinical evidence, including urinary miRNAs and isoprostanes. Genetic alterations frequently identified in PCa, including BRCA1/BRCA2, ETS gene fusions, and AR changes, are also discussed, offering insights into risk assessment and precision treatment strategies. By evaluating the latest developments and applications of PCa biomarkers, this review contributes to an enhanced understanding of their role in disease management.

## 1. Introduction

Prostate cancer (PCa) is the second most prevalent cancer and the fifth leading cause of cancer-related fatalities in men worldwide. In 2020, approximately 1,414,259 new cases of PCa were reported worldwide, resulting in 375,304 PCa-related fatalities [1]. According to recent data from the U.S. SEER database, the estimated new cases of PCa are 288,300 in the United States in 2023, accounting for 14.7% of all newly identified cases during this period. The estimated 5-year survival rate of these patients is 97.1% [2].

PCa is highly variable, with about 20% of cases being high-risk [3]. Tailored treatments range from active surveillance to intensive therapies [4,5]. An accurate risk assessment is critical for treatment decisions [6]. The diagnosis relies on methods such as a digital rectal examination (DRE), a prostate-specific antigen (PSA) assessment, imaging, and a tissue biopsy with Gleason grading [5]. Gleason scores of 1–5 usually indicate benign tumors; scores of 6–7 indicate tumors that are manageable; and scores of 8–10 signal advanced disease [7].

The current diagnostic methods for PCa have various limitations that can lead to overdiagnoses and overtreatment [8]. For example, PSA testing, while aiding early detection, lacks precision (20–40% accuracy), as non-malignant conditions can cause PSA elevation, potentially resulting in false positives [9,10]. Conversely, slow-growing PCa with normal PSA levels may be missed, increasing the mortality risk [11]. To address these issues, a biopsy may be suggested when two abnormal PSA levels or a palpable abnormality are present [12]. Low-risk individuals with Gleason scores of 6 or lower and PSA levels below 10 μg/L may opt for active surveillance to avoid unnecessary treatment until the disease progression necessitates intervention [13,14].

In recent years, the growing understanding of the malignant biological characteristics of PCa and its molecular attributes has enabled the discovery of multiple biomarkers, which have been integrated with current diagnostic methods, risk assessments, and treatment selections. Biomarkers guide treatment decisions by identifying which therapies are most likely to be effective for a specific patient. This prevents overtreatment and minimizes the risk of adverse effects. A personalized treatment based on biomarker profiles can lead to improved patient outcomes, including higher response rates, longer survival, and a better quality of life. This review comprehensively explores biomarkers throughout the clinical continuum of PCa, from diagnosis to treatment. It encompasses both well-established clinical biomarkers and emerging ones that require further validation.

## 2. Biomarker for PCa Diagnosis

The specificity of using PSA as a screening tool for PCa has its limitations, especially when the PSA levels are lower than 10 μg/L. This lack of specificity results in a substantial proportion of men undergoing biopsies, either to confirm or rule out a malignancy, unnecessarily. Remarkably, studies have shown that 65% to 75% of men with PSA levels between 3/4 and 10 μg/L do not exhibit any signs of detectable PCa through a biopsy [15]. To improve precision and decrease the occurrence of unnecessary or repeated biopsies, various supplementary tests such as the PHI, the 4K score, and PCA3 have been proposed.

The basis for the development of numerous tests is that PSA can manifest in various forms [16,17]. Initial investigations revealed distinct PSA forms within blood, which were mainly discovered in association with serum protease inhibitors—particularly α1-antichymotrypsin—accounting for around 70% to 90% of its composition. Approximately 10% to 30% of PSA remains unbound [16,17]. This unbound PSA in serum contains three principal forms: pro-PSA, BPSA, and intact PSA. Significantly, pro-PSA can manifest in various ways, including the native proPSA form, which harbors a 7-amino-acid pro-peptide leader [(−7) proPSA], along with versions presenting truncated pro-peptide leader sequences. These truncated proPSA iterations include proPSA with a 5-amino-acid (−5) proPSA, 4-amino-acid (−4) proPSA, or 2-amino-acid (−2) proPSA, among others [18,19].

### 2.1. The Prostate Health Index (PHI)

The calculation of the prostate health index (PHI) involves measuring −2proPSA, the percentage of free PSA (fPSA), and the total PSA (tPSA). These values are combined using the formula (−2 proPSA/fPSA) × √tPSA provided by Beckman Coulter, Inc. (Brea, CA, USA) [20]. The PHI has shown exceptional proficiency in the examination of PCa, including more aggressive subtypes, surpassing both the total PSA and the percentage of free PSA [20,21,22,23]. In a meta-analysis of 2919 patients across eight studies, the PHI exhibited a combined sensitivity of 90% and a combined specificity of 31.6% with regard to PCa detection [23]. The PHI consistently outperformed the total PSA and percentage of free PSA in terms of accurate PCa detection, especially in men with PSA levels between 2 μg/L and 10 μg/L. Additionally, a 6-year follow-up study indicated that the initial PHI levels could predict the 6-year risk of PCa, with a higher PHI (>35) indicating a greater risk that would prompt closer monitoring [24]. Furthermore, combining the PHI with other diagnostic methods, such as multiparametric magnetic resonance imaging (mpMRI), improved its performance. In an Asian population, the combination of the PHI and mpMRI demonstrated a higher accuracy in detecting clinically significant PCa (csPC) compared to the PHI or mpMRI alone, with an AUC of 0.873 versus 0.735 (*p* = 0.002) and 0.873 versus 0.830 (*p* = 0.035), respectively [25].

In clinical settings, the PHI is primarily used to improve individual risk assessments for early PCa detection [26] and reduce unnecessary biopsies in males with borderline PSA levels, resulting in reductions that range from approximately 15% to 45%, depending on the selected threshold [27]. Nevertheless, this evasion may result in a small percentage of overlooked cancers—typically less than 10% when the threshold is established at 25 [27]. The ideal pre-analytical treatment and storage conditions for PHI measurements have yet to be determined. Further research is needed to ascertain whether plasma or serum is the preferred matrix for precise measurements. A recent multicenter study also showed that the PHI reference ranges need to be adjusted for different populations [28].

### 2.2. 4K Score

The 4K score tests include evaluating the levels of total PSA, free PSA, intact PSA (a form of free PSA), and human kallikrein 2 (hK2) [29]. An algorithm combines the levels of these biomarkers with the patient’s age, the results of a DRE, and any previous biopsy outcomes to predict the probability of an individual having high-grade PCa. Numerous studies have shown that, like the PHI, the 4K score is more accurate for diagnosing PCa in general and high-grade PCa specifically, compared to PSA or the percentage of free PSA [30,31,32,33,34]. For example, the AUC for the 4K score in the ProtecT trial, which included 4765 patients, was 0.719, whereas it was 0.634 for PSA in all cancers and 0.820 compared to 0.738 for high-grade PCa [30]. A meta-analysis of published studies found that the 4K score and the PHI showed a similar diagnostic accuracy for high-grade PCa [34]. Similar to the percentage of free PSA and the PHI, the 4K score has a significant clinical benefit, as it helps patients avoid unnecessary biopsies. Reported reductions in this magnitude have been observed, varying between 41% and 57% depending on the selected threshold [27]. Like the PHI, this decrease in the 4K score might overlook a small proportion of clinically significant PCa [35]. Another issue arises as a result of the variability in the cutoff values for the 4K score across different studies, leading to heterogeneity in PCa diagnoses. A recent meta-analysis suggested that a 4K score below 7.5% signifies a low risk, whereas cutoff values from 7.5% to 10% provide a high level of accuracy for a high-grade PCa diagnosis. Nevertheless, larger-scale studies are needed to confirm this finding [36].

### 2.3. PCA3

PCA3, also known as DD3, is a prostate-specific mRNA biomarker that has promising potential for the detection of prostate cancer (PCa). Studies noted significantly higher PCA3 levels (10 to 100 times) in 53 out of 56 PCa tissue samples compared to adjacent non-cancerous prostate tissue. PCA3 was absent in non-prostatic tissues, but present in normal prostates and benign prostate hypertrophy [37]. The Progensa PCA3 assay, which detects both PCA3 and PSA mRNA in urine samples after a digital rectal examination, has diagnostic potential [38]. The pooled data from 46 studies involving 12,295 individuals showed a promising sensitivity (0.65) and specificity (0.73) for PCa diagnoses, with an AUC of 0.75 [39].

Although PCA3 is not expected to replace PSA as the main indicator for PCa, the combination of their measurements could greatly improve the accuracy of PCa diagnoses. PCA3 testing could be especially beneficial for individuals with high PSA levels and biopsies that show no abnormalities histologically [40]. In such cases, PCA3 can be used to determine whether a repeat biopsy is necessary. The FDA approved the use of the PROGENSA PCA3 test in 2012, in conjunction with other patient data, to help decide if men aged ≥50 years, who have had one or more negative prostate biopsies in the past and are recommended for a repeat biopsy by a urologist according to current practice standards, actually need said biopsy. However, PCA3 has certain limitations. Firstly, there is controversy surrounding the optimal thresholds for PCA3. Various studies have employed different PCA3 score thresholds, with some utilizing a threshold of ≥35, while others favor a threshold of <35. Recent investigations have indicated that a PCA3 score of 35 strikes an optimal balance between sensitivity and specificity in diagnosing PCa, whereas a PCA3 score lower than 25 may predict the presence of pathological indolent PCa. Secondly, PCA3, as an mRNA, is inherently unstable, necessitating precise and accurate handling and preservation methods [41].

### 2.4. Mi-Prostate Score (MiPS)

The Mi-prostate score (MiPS) is a predictive algorithm that includes serum PSA and the urinary biomarkers PCA3 and TMPRSS2:ERG. There is evidence suggesting that it is markedly superior to PSA for diagnostic purposes [27,42]. Significantly, the MiPS has demonstrated the capability not just of detecting the existence of PCa before a biopsy, resulting in a significant reduction of 35–47% in unnecessary biopsies and overdiagnoses, but also of predicting high-grade PCa on a biopsy, making it a valuable instrument for estimating individual risk [43]. However, the MiPS has several obvious shortcomings: Firstly, it demands a high level of technical expertise and platform capability for effective implementation. Secondly, it presents an important limitation stemming from variations in the prevalence of the TMPRSS2:ERG gene fusion among different racial groups. Research has demonstrated substantial differences, with this fusion being present in 50% of Caucasians, 31.3% of African Americans, and 15.9% of Japanese patients. The potential implications of this variance with regard to the applicability of the MiPS in non-Western patient populations remain uncertain and should be considered by healthcare providers [44].

### 2.5. Urinary miRNAs (umiRNAs)

Urinary miRNAs have emerged as promising biomarkers for detecting prostate cancer (PCa), offering valuable tools to distinguish between malignant and benign tumors. Elevated levels of urinary miR-100 and miR-200b have been associated with advanced PCa, while miR-196a-5p and miR-501-3p are downregulated in urinary exosomes, holding promise as PCa biomarkers. Notably, the levels of miR-21, miR-141, and miR-375 were significantly higher in the urinary samples of PCa patients compared to those of healthy individuals. Furthermore, an increased expression of miR-141 was observed in patients with higher Gleason scores. The upregulation of miR-21-5p, miR-141-3p, and miR-205p in urine samples exhibited a higher specificity for PCa detection compared to traditional PSA testing. Conversely, miR-19b and miR-26a showed significant downregulation, while miR-320a was upregulated in PCa patients, supporting their potential as valuable biomarkers.

Based on miRNA research, a ratio analysis was employed to enhance the diagnostic accuracy. The urinary miR-1913 to miR-3659 ratio was found to be elevated in PCa patients, particularly benefiting those with total serum PSA levels between 3 and 10 ng/mL. Additionally, the expression ratio of urinary miR-H9 to miR-3659 was significantly higher in the PCa group. While urinary miRNAs show promise as supplemental biomarkers to complement PSA testing in PCa diagnoses, none have yet been officially recognized as standalone diagnostic markers [45].

Table 1 lists additional reflex biomarkers that may aid in making informed decisions about biopsies for men with borderline PSA levels.

## 3. Biomarker for PCa Risk Stratification (Prognosis)

Table 2 presents a variety of tissue biomarker assays that have emerged in the contemporary period to evaluate the severity and predict the outlook for individuals diagnosed with PCa. Notably, certain tests are multi-analyte in nature, measuring various molecular entities, particularly mRNAs. Decipher, Oncotype DX (Prostate), and Prolaris are the tests with the highest level of substantiation among those listed in Table 2. The main purpose of these tests is to provide information on tumor aggressiveness and patient outcomes, even though they have been evaluated in various scenarios and used at different endpoints.

### 3.1. Decipher

Decipher™, a genomic classifier developed collaboratively by GenomeDx Biosciences based in Vancouver, BC, Canada, and Mayo Clinic, is designed to forecast the probability of metastasis after a radical prostatectomy (RP). The foundation of this tool relies on the examination of 1.4 million genetic markers, including coding and non-coding RNA. The signature is formed by 22 RNA biomarkers that affect various biological signaling pathways such as cell differentiation, proliferation, structure, adhesion, motility, immune response, cell-cycle progression, and androgen signaling [57]. The test produces a genomic classifier (GC) that assigns a continuous risk score between 0 and 1. The risk of metastasis within 5 years after surgery is indicated by each score, with a score of 1 representing the highest probability. Decipher™ has received approval in the United States for evaluating the likelihood of experiencing biochemical recurrence (BCR) or clinical progression (e.g., metastasis) in post-RP PCa patients with an adverse pathology (pT3 and/or positive margins or biochemical failure) or PSA persistence or recurrence during the follow-up period.

The role of Decipher™ in the classification and personalized treatment of prostate cancer has been demonstrated. A systemic review found that the GC is an independent prognostic factor for various critical endpoints, including biochemical recurrence, metastasis, and cancer-specific survival [67]. One of the most important utilities of the GC is its ability to classify the high risk of aggressive disease in the patients who received an RP. A prospective study revealed that, among high-risk GC patients, those who underwent adjuvant RT(ART) had a lower 2-year PSA recurrence compared to those who did not receive ART (3% vs. 25%, *p* = 0.013). However, no difference in PSA recurrence was observed between ART and no ART in low-/intermediate-risk patients [68]. Furthermore, in the ancillary study of the NRG/RTOG 9601, the GC score was associated with a distant metastasis (DM) and prostate-cancer-specific mortality (PCSM) after a prostatectomy. For patients with low PSA (<0.7 ng/mL) and a low GC, there may be little clinical benefit from adding bicalutamide to salvage radiotherapy [69]. Based on this evidence, the NCCN guidelines strongly recommend that patients with a high Decipher score (GC > 0.6) receive intensive treatment (EBRT + ADT) if adverse features or PSA persistence/recurrence are detected after an RP [5].

However, to truly assess its value, it is important to incorporate extensive clinical implementation and long-term data, despite the recommendations outlined in the National Comprehensive Cancer Network (NCCN) guidelines for its use [5,70].

### 3.2. Prolaris

Prolaris^®^, a predictive examination model developed by Myriad Genetics (located in Salt Lake City, UT, USA), employs the manifestation of 31 genes related to cell-cycle progression (CCP) and 15 genes associated with housekeeping (shown in Table 2). This test was initially developed as an expansion of their breast cancer trials. A CCP score [71] is produced, indicating a proliferative index. This may improve predictive abilities compared to the clinicopathological factor. Similar to other tests, this assessment predicts the probability of future occurrences, such as the BCR and mortality specific to PCa, rather than precise treatment recommendations.

The CCP score has prognostic value in various clinical settings, as it can be tested in diverse sample types such as biopsies, a transurethral resection of the prostate (TUR-P), and RPs. A retrospective study found that the CCP score was useful for predicting biochemical recurrence after a prostatectomy, as well as the time to PCa-related death after a TUR-P [71]. In a conservative cohort of localized PCa patients diagnosed via a needle biopsy, the CCP score remained a significant predictor (HR 1.76) independent of clinical variables. Another validation cohort showed that a predefined CCP score could predict 10-year prostate-specific death and identify the low-risk group that did not require radical treatment [72]. Many studies have utilized the CCP score to predict the aggressive potential of PCa. A systemic review demonstrated that the CCP score could have an impact on clinical decisions and reduce unnecessary surgeries for suitable low-risk patients [73]. This tool is employed to help determine the optimal approach between immediate and delayed (conservative management) treatment for males with indolent cancer.

In accordance with these findings, NCCN recommends utilizing CCP after a biopsy for NCCN cases categorized as very low-, low-, or favorable–intermediate-risk PCa during the diagnostic process for individuals with a projected lifespan of at least 10 years [5].

### 3.3. Oncotype DX^®^

Genomic Health (Redwood City, CA, USA) developed Oncotype DX^®^, a widely recognized test that can predict and forecast outcomes in breast cancer patients. This test has since been adapted for PCa. This test uses quantitative RT-PCR on FFPE tissue obtained from needle biopsies to analyze the expression of 12 genes associated with cancer across four biological pathways (stromal response, androgen signaling, proliferation, and cellular organization), along with five reference genes. The GPS (genomic prostate score) is calculated by algorithmically combining these components [61]. The primary purpose of this tool is to help patients make informed decisions about the most appropriate treatment approach (active monitoring versus treatment) after being diagnosed with low- or low–intermediate-risk PCa. For these patients, a higher genomic prostate score (GPS) on a scale from 0 to 100 corresponds to an increased likelihood of an adverse pathology during a radical prostatectomy (RP).

The Oncotype DX^®^ has demonstrated its specific predictive value and impact on clinical decisions. In a diverse cohort of very low- to intermediate-risk PCa, the GPS was found to be associated with the time to BCR, the time to metastasis, and the presence of adverse pathological features (primary Gleason pattern 4 or any pattern 5 and/or pT3) after adjusting the NCCN risk group. This association was the same for both African American and Caucasian patients [74]. The information provided by the GPS may lead to an increased physician’s recommendation of active surveillance in very low- or low-risk groups [75]. Unlike other biomarkers that are only associated with long-term clinical outcomes, the GPS may predict an adverse tumor pathology (AP) (extraprostatic extension, positive surgical margin, and seminal vesicle invasion) after an RP [76]. Accordingly, the Oncotype DX^®^ GPS would provide valuable references for counseling patients about radical surgery or other options. Furthermore, a recent study also showed that the GPS is an independent prognostic factor for clinical outcomes in patients with localized PCa, similar to a radical prostatectomy [77].

The NCCN guidelines specifically recommend using Oncotype DX^®^ to make decisions after a biopsy in cases of a localized disease at diagnosis, in patients with a life expectancy of ≥10 years [5].

Although the current biomarkers for risk stratification in PCa are primarily identified in tissue samples obtained from surgical procedures or biopsies, emerging urinary biomarkers also warrant attention. A study indicated that individuals with PCa exhibit higher levels of urinary 8-hydroxy-2-deoxyguanosine (8-OHdG) and 8-iso-prostaglandin F2α (8-IsoF2α) than healthy subjects. However, a robot-assisted radical prostatectomy (RARP) could normalize these oxidative stress markers [78]. The measurement of these urinary biomarkers holds a potential role to assess the radicality of treatment and the aggressive risk in PCa patients.

## 4. Genetic Biomarkers for PCa Treatment

The development of PCa is significantly influenced by familial history, which is a firmly established risk factor. Some families exhibit such a strong genetic transmission that the hereditary pattern emulates autosomal dominance traits [79]. Although there are suggestions that environmental factors may contribute to an overestimation of the hereditary influence in PCa, it is important to note that genetic predisposition plays a crucial role and remains significant even when environmental factors are taken into account [80,81]. Approximately 100 susceptibility loci have been identified through genome-wide association studies, which collectively contribute to approximately 39% of the familial risk associated with the occurrence of PCa [79]. Examining genetic markers associated with these loci in people with a familial background of PCa could not only assist with risk assessment and stratification, but could also allow for earlier detection and timely medical interventions.

The most frequently identified genetic changes in PCa are as follows.

### 4.1. BRCA1/BRCA2

Mutations in the BRCA1 and BRCA2 genes [82,83,84] are the strongest indicators of PCa. BRCA2 mutations have been strongly associated with an increased susceptibility to developing PCa (2.5–8.6 times higher by age 65), as well as an earlier onset of the disease, aggressive tumor growth, metastases, and lower survival rates [85,86,87,88]. mCRPC patients with BRCA2 mutations have shown notable genomic instability in the MED12L/MED12 gene axis, which is closely related to metastatic disease. This explains the unique aggressiveness of tumors containing these genetic abnormalities [89]. Recently, PARP inhibitors such as rucaparib and olaparib, which have been approved by the FDA, have proven capable of hindering the DNA damage response in PCa with BRCA1 and/or BRCA2 mutations. This effectively delays the progression of cancer in refractory or metastatic situations [90,91].

### 4.2. ETS Gene Fusions

Approximately 50% of PCa patients of Caucasian descent have detectable ETS gene fusions [92,93]. The fusion of androgen-regulated transmembrane protease, serine 2 (TMPRSS2) with ERG (TMPRSS2:ERG) is the most frequently observed. The MiPS test includes this unique PCa fusion, offering diagnostic, prognostic, and risk-assessment benefits, as well as a possible therapeutic target [94,95,96,97]. More than 50% of PCa patients exhibit an increase in ERG gene expression, predominantly characterized by the presence of TMPRSS2:ERG fusion or other TMPRSS2 fusion partners such as ETV1, ETV4, ETV5, and FLI1 [97,98]. Besides TMPRSS2, other androgen-responsive 5′ partners combined with ERG consist of SLC45A3, HERPUD1, and NDRG1 [99,100,101]. It is believed that these combinations arise from continuously activated AR signaling, causing intronic AR binding. This unites the translocation loci and triggers site-specific, double-stranded breaks through genotoxic stress and specific stress-induced enzymatic processes [102]. Interestingly, the occurrence of ERG fusion varies among different races: Caucasian Americans (CAs) exhibit ERG expression nearly twice as frequently as African Americans (AAs) [103,104]. This racial variation may have a substantial impact on precision therapeutic approaches, especially when considering the observed racial differences with regard to the effectiveness of different PCa treatments [105].

### 4.3. Androgen Receptor (AR)

The AR is a transcription factor that controls the growth and development of the prostate. Mutations in its ligand-binding domain contribute to hormone-sensitive PCa (HSPC) and mCRPC, accompanied by other AR changes such as amplification and structural rearrangements [92,93]. Contrarily, hormone-naïve PCa rarely exhibits AR alterations [106]. Intriguingly, AR mutations are clinically relevant in 34% of CRPC patients, hinting at their potential responsiveness to AR-targeted therapies [107]. Nevertheless, additional repetitive modifications may arise during the recurrence of CRPC, leading to resistance against ADT. Examples include the FOXA1, SPOP, and IDH1 gene mutations, and the NCOA2 gene gain [93,106,107]. The AR splice variant 7 (AR-V7) has a deletion in exon 7. This deletion causes the ligand-binding domain to be absent, making it constitutively active and causing AR resistance [108]. AR-V7 is linked to CRPC development and is associated with reduced survival in enzalutamide- or abiraterone-treated patients [109,110]. Notably, patients with a positive AR-V7 status who are undergoing taxane treatment demonstrate improved survival, suggesting that AR-V7 has the potential to forecast the response to therapy in mCRPC [111,112,113]. The guideline from AUA/ASTRO/SUO highlights AR-V7 as a potential indicator for forecasting various systemic treatment reactions and improving long-term outcomes [114,115].

## 5. Conclusions

Multiple biomarkers have been identified as relevant to the diagnosis, risk prognosis, and treatment selection of PCa. This review lists biomarkers that have been integrated into relevant clinical guidelines. Most of these emerging biomarkers have demonstrable value with regard to reducing overdiagnoses in indolent disease, identifying a high-risk-of-aggressive group, and providing individual treatment to patients with a genetic predisposition. Meanwhile, ongoing clinical trials continue to emphasize the importance of carefully selecting suitable populations to provide more prospective data for the application of these tools.

Although the presence of several biomarkers has provided more information beyond regular clinical factors, identifying the optimal utility of PCa biomarkers remains challenging. Firstly, the procedure of identifying these molecular classification instruments is intricate, necessitating an excellent sample quality. It is imperative to streamline the processes of testing and decrease the expenses associated with testing. In addition, a well-designed combination of approaches and endpoints during clinical trials may hold the potential to achieve an accurate molecular risk stratification for PCa patients in the future. Moreover, the quickly growing range of biomarkers accessible for therapeutic targeting indicates the upcoming arrival of precision therapies that can help larger groups of patients. The trajectory may lead to significant progress in the treatment of metastatic or refractory conditions, leading to better outcomes and longer survival for heterogeneous groups of PCa patients.

## Figures and Tables

**Table 1 diagnostics-13-03350-t001:** Advancements in the development of diagnostic models for PCa.

Detection Assay	Analytes	Clinical Utility	Limitations
4K score [35,46]	Serum tPSA, fPSA, HK2, and intact PSA; a DRE; and a history of a negative biopsy	Mainly used as a diagnostic marker; the EAU guidelines recommend using it for patients with PSA levels between 2 and 10 ng/mL. The NCCN guidelines recommend using it for patients with negative biopsies. It is used to diagnose PCa with a Gleason score (GS) of ≥7, and some studies have used it to predict the risk of a distant metastasis.	The cutoff values in different guidelines have not been determined. It would miss a small amount of csPCa.
PHI [47,48]	−2 proPSA/fPSA × √tPSA	Mainly used as a diagnostic biomarker; the EAU guidelines recommend stratifying patients with PSA levels between 2 and 10 ng/mL to reduce biopsies. The AUA guidelines recommend it as a second-line monitoring tool. The FDA approved the PHI for the early detection of Pca.	Occasionally leads to missed cancers. Optimal pre-analytical handling and sample storage conditions have yet to be determined.
SelectMDx [49,50]	mRNA expression of HOXC6, DLX1, and KLK3 in urine samples following a DRE, combined with the family history, age, a DRE, and PSA	Mainly used as a diagnostic biomarker; it assesses the associated risk in patients with PCa and a GS ≥ 7.	The clinical application of this test is currently under evaluation by the NCCN expert committee.
ConfirmMDx [51,52]	The epigenetic modifications of three genes (GSTP1, APC, and RASSF1) in prostate tissue	Can be applied in situations where tumor tissue cannot be obtained through a biopsy, as recommended by NCCN and EAU guidelines.	The FDA has not yet approved it, and large-scale clinical applications have not been initiated or validated.
Progensa^®^ PCA3 [37,41]	PCA3 (non-coding RNA)	NCCN and EAU guidelines recommend using it after confirmation of negative biopsy results. In addition, this score can predict patients with PCa and a GS ≥ 7.	The thresholds remain controversial. PCA3 is unstable and needs more effort to capture and preserve.
Mi-prostate score [43,53]	Serum PSA; urinary PCA3 and TMPRSS2:ERG	Used for the early detection of invasive PCa.	It requires a relatively high level of technical platform.
The Stockholm3-test (sthlm3 model) [54]	Clinical variables (age, family history, and biopsy results), serum markers (tPSA, fPSA, hK2, MIC1, and MSMB), and 254 single-nucleotide polymorphisms (SNPs) of the HOXB13 gene	It is used for detecting highly malignant PCa (GS ≥ 7).	It is only applicable in Nordic countries (Sweden, Norway, Denmark, Finland, etc.). The scope of SNP information is relatively limited, and the population is relatively homogeneous.
ExoDx prostate IntelliScore [55]	Exosomal RNA levels of PCA3, ERG, and SPDEF from non-DRE urine	The NCCN guidelines recommend it as the preferred option for initial or repeat biopsies; used to detect high-grade PCa (GS ≥ 7).	There is currently a lack of standardized and executable protocols for extracellular vesicle isolation and detection.
ERSPC risk calculators (RC) [56]	Family history, age, urinary system symptoms, tPSA level, a DRE, the prostate volume, multiparametric MRI imaging data, and the biopsy history	The EAU guidelines recommend it for assessing the risk of PCa.	-

**Table 2 diagnostics-13-03350-t002:** Research progress on predictive models for PCa prognosis.

Detection Assay	Analytes	Clinical Utility	Limitations
Decipher genomic classifier [57]	Assessing 22 lncRNAs in PCa tissue specimens on a scale from 0 to 1 to evaluate the risk.	NCCN guidelines recommend its use for risk stratification after a radical prostatectomy. It is also used to guide postoperative radiation therapy and ADT.	Comprehensive clinical applications and long-term data are still needed.
Prolaris (CCP score) [58,59,60]	Detecting 31 cell-cycle-related genes and 15 housekeeping genes in prostate tissue.	NCCN guidelines recommend using it to help assess the prognosis and risk stratification in patients with or without treatment. Some studies also use it to identify indolent cancer.	The test was not trained for a specific a priori endpoint. There are currently no relevant prospective randomized controlled trials to validate the efficacy of this model.
Oncotype DX^®^ GPS [61]	Using PCa tissue samples to detect the expression of 17 genes involving 4 pathways: stromal response, androgen signaling, proliferation, and cellular organization.	NCCN guidelines recommend its use to help assess patients’ prognosis and risk stratification. It has specific value in the prediction of an adverse pathology after an RP.	-
ProMark [62]	Using immunofluorescence to detect the expression of 8 protein molecules in PCa tissue, graded from 0 to 1.	NCCN guidelines recommend its use for prognostic risk stratification.	There are currently no relevant prospective randomized controlled trials to validate the predictive power of this model.
ADT-RS [63]	Screening for 49 relevant genes from the Decipher GRID database to predict the response to ADT (androgen deprivation therapy).	A higher ADT-RS (androgen deprivation therapy response score) indicates a greater benefit from ADT treatment; patients with higher ADT-RSs experienced a decrease in the distant metastasis rate after ADT.	There is a lack of multicenter, prospective data to confirm these findings.
PAM50 [64,65]	By detecting the expression of 50 genes (PAM50) and 5 control genes in surgical specimens, PCa is classified into molecular subtypes: Lum A, Lum B, or basal.	The luminal B subtype benefits from postoperative ADT, while the benefits for the other subtypes are not clear. Recent studies have shown the potential application of molecular subtyping for patients with mCRPC (metastatic castration-resistant PCa).	Its predictive efficacy has not been confirmed in further clinical trials.
RSI [66]	A total of 11 relevant genes were selected from the molecular expression profiles of more than 60 irradiated cell strains.	It can predict the sensitivity of PCa patients to radiation therapy, but cannot predict the patient’s outcome after radiation therapy.	Its efficacy lacks confirmation from clinical trials.

## Data Availability

No new data were created or analyzed in this study. Data sharing is not applicable to this article.

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
