# Peer review of "Biomarkers for Prostate Cancer: From Diagnosis to Treatment"

_diagnostics, 2023, doi:10.3390/diagnostics13213350_

Round 1

Reviewer 1 Report

Dear Authors,

Congratulations on your manuscript titled "Biomarkers for Prostate Cancer: From Diagnosis to Treatment."!

I have carefully reviewed the sections you provided and would like to offer some constructive comments.

Positive Feedback:

Relevance and Importance: Your manuscript effectively highlights the global significance of prostate cancer, emphasizing its prevalence and impact on cancer-related fatalities in men. This establishes the importance of the research topic.

Clarity and Informative: The introduction section provides clear statistics and data, making it easy for readers to understand the current status of prostate cancer globally and in the United States. It sets the stage for the importance of biomarkers.

In-Depth Overview: The introduction provides a comprehensive overview of the heterogeneity of prostate cancer, emphasizing the need for accurate risk stratification and personalized treatment. It also discusses current diagnostic methods and their limitations.

Transition to Biomarkers: Your manuscript smoothly transitions into the discussion of biomarkers, highlighting the limitations of current methods and the need for new diagnostic tools.

Logical Flow: The manuscript follows a logical flow from the general introduction to specific biomarkers, prognosis, and treatment options, which aids in reader comprehension.

However, minor changes are necessary. Please find some recommendations in the following part of my review.

Introduction section: While your introduction provides a comprehensive overview, it is quite lengthy and contains a lot of information. Consider condensing some parts for improved readability.

Complex Sentences: Some sentences are complex and might require readers to re-read for complete understanding. Simplifying certain sentences could enhance clarity. Please find some examples below.

Original "The latest statistics on global cancer epidemiology indicate that there were around 1,414,259 new instances of PCa and 375,304 fatalities related to PCa across the world in 2020 [1]."

Simplified: "In 2020, global cancer statistics reported approximately 1,414,259 new cases of PCa worldwide, resulting in 375,304 PCa-related fatalities [1]."

Original: "Nevertheless, even with the presence of these suggestions outlined in the guidelines of the National Comprehensive Cancer Network (NCCN) [5], a thorough evaluation of its actual worth requires the incorporation of extensive clinical implementation and long-term data [53]."

Simplified: "However, to truly assess its value, it's important to incorporate extensive clinical implementation and long-term data, despite the recommendations outlined in the National Comprehensive Cancer Network (NCCN) guidelines [5] for its use [53]."

Original: "Although there are difficulties associated with identifying clinically useful biomarkers for PCa, the incorporation of these biomarkers into well-designed combination approaches during clinical trials has the potential to greatly improve the accuracy of diagnosis, prognostic forecasts, and treatment effectiveness."

Simplified: "While challenges exist in identifying clinically useful PCa biomarkers, integrating them into well-designed combination approaches during clinical trials holds the potential to significantly enhance the accuracy of diagnosis, prognostic predictions, and treatment outcomes."

Conclusion Section: It is important to ensure that the manuscript has a clear and concise conclusion section that summarizes the main points discussed in the introduction and subsequent sections. A shortening of this section is also necessary

References Section: Please review all refferences in the mansucript, as it does not apply to the format recommended in the Instructions for authors section.

Your manuscript shows promise, and I believe that with some revisions and attention to the mentioned areas, it will be a valuable contribution to the field of prostate cancer research. I look forward to reviewing the complete manuscript when it becomes available.

Author Response

Thank you very much for taking time to review this manuscript. We have carefully reviewed the comments and have revised the manuscript accordingly. Please see the attachment.
Please find the detailed responses and the revised portions are marked in red in the manuscript.

Thank you very much once again.

Reviewer 2 Report

This study aims to provide an overview of biomarkers for the identification, prediction, and treatment of PCa.

I believe that the study has sufficient merit to be considered for publication on Diagnostics, although major revisions are required.

MAJOR COMMENTS

-       The abstract should provide a concise summary of the key findings and contributions of the review. It should briefly mention the biomarkers discussed and their clinical significance.

-       Keywords: Consider adding specific keywords related to the biomarkers discussed in the article to improve discoverability.

-       Introduction: Include a brief overview of the current challenges in prostate cancer diagnosis and treatment to set the context for the article.

Clarify the scope of the review, such as whether it focuses on specific biomarkers or covers a broader range.

Consider adding a statement about the importance of biomarkers in personalized medicine and their potential impact on patient outcomes.

-       Biomarker for PCa Diagnosis: When discussing specific biomarkers like PHI, 4K score, PCA3, and MiPS, provide more recent research findings, including clinical trial results and their implications for diagnosis. Mention any limitations or challenges associated with the use of these biomarkers in clinical practice. The authors should add a paragraph on microRNAs, emerging biomarkers for the detection and prognosis of prostate cancer. In this regard, I recommend this reference that I think is important and that can be of great help when modifying the manuscript (doi: 10.3390/ijms241310846, PMID 37446024).

-       Biomarker for PCa Risk Stratification (Prognosis): Discuss the clinical utility of Decipher, Prolaris, and Oncotype DX in more detail. Provide insights into how I these tests impact treatment decisions and patient outcomes. The authors should better discuss the role of emerging biomarkers of the radicality of treatment in patients with PCa. ( 10.3390/jcm11206102).

-       Conclusion: Summarize the key takeaways from the article, emphasizing the clinical implications of the discussed biomarkers.

-       Provide insights into future research directions and the potential impact of emerging biomarkers on prostate cancer management.

Moderate editing of English language required

Author Response

(The authors gave the same response as above.)

Round 2

Reviewer 1 Report

Congratulations. I support the publication of the manusceipt in the current form.